# In Vitro Assessment of Fluoropyrimidine-Metabolizing Enzymes: Dihydropyrimidine Dehydrogenase, Dihydropyrimidinase, and β-Ureidopropionase

**DOI:** 10.3390/jcm9082342

**Published:** 2020-07-22

**Authors:** Eiji Hishinuma, Evelyn Gutiérrez Rico, Masahiro Hiratsuka

**Affiliations:** 1Laboratory of Pharmacotherapy of Life-Style Related Diseases, Graduate School of Pharmaceutical Sciences, Tohoku University, Sendai 980-8578, Japan; ehishi@ingem.oas.tohoku.ac.jp (E.H.); gutierrez@tohoku.ac.jp (E.G.R.); 2Tohoku Medical Megabank Organization, Tohoku University, Sendai 980-8573, Japan; 3Advanced Research Center for Innovations in Next-Generation Medicine, Tohoku University, Sendai 980-8573, Japan; 4Department of Pharmaceutical Sciences, Tohoku University Hospital, Sendai 980-8574, Japan

**Keywords:** fluoropyrimidine, dihydropyrimidine dehydrogenase, dihydropyrimidinase, β-ureidopropionase, genetic polymorphism

## Abstract

Fluoropyrimidine drugs (FPs), including 5-fluorouracil, tegafur, capecitabine, and doxifluridine, are among the most widely used anticancer agents in the treatment of solid tumors. However, severe toxicity occurs in approximately 30% of patients following FP administration, emphasizing the importance of predicting the risk of acute toxicity before treatment. Three metabolic enzymes, dihydropyrimidine dehydrogenase (DPD), dihydropyrimidinase (DHP), and β-ureidopropionase (β-UP), degrade FPs; hence, deficiencies in these enzymes, arising from genetic polymorphisms, are involved in severe FP-related toxicity, although the effect of these polymorphisms on in vivo enzymatic activity has not been clarified. Furthermore, the clinical usefulness of current methods for predicting in vivo activity, such as pyrimidine concentrations in blood or urine, is unknown. In vitro tests have been established as advantageous for predicting the in vivo activity of enzyme variants. This is due to several studies that evaluated FP activities after enzyme metabolism using transient expression systems in *Escherichia coli* or mammalian cells; however, there are no comparative reports of these results. Thus, in this review, we summarized the results of in vitro analyses involving DPD, DHP, and β-UP in an attempt to encourage further comparative studies using these drug types and to aid in the elucidation of their underlying mechanisms.

## 1. Introduction

Fluoropyrimidine drugs (FPs), including 5-fluorouracil (5-FU) and its oral prodrugs tegafur, capecitabine, and doxifluridine, are widely used in the treatment of solid tumors in the gastrointestinal tract, breast, liver, lung, head, and neck [1,2,3]. FP-based treatments have a narrow therapeutic index, which has led to severe adverse effects in approximately 30% of cancer patients, including mucositis, diarrhea, neutropenia, thrombocytopenia, and hand–foot syndrome [4,5,6,7,8]. Additionally, severe treatment toxicities could lead to treatment interruption, which increases the subsequent risk of therapeutic failure as well as patient death [9].

Genetic polymorphisms of thymidylate synthase (TYMS), methylene tetrahydrofolate reductase (MTHFR), and miR-27a are associated with the development of severe toxicities as well as treatment resistance; however, FP-related toxicity is mainly dependent on FP catabolism. Over 80% of an administered dose of 5-FU is rapidly degraded by three consecutive enzymes belonging to the endogenous pyrimidine, uracil, and thymine catabolic pathways (Figure 1), the only known 5-FU in vivo degradation pathway. Initially, the rate-limiting enzyme dihydropyrimidine dehydrogenase (DPD, EC 1.3.1.2), mainly found in the liver, catalyzes the reduction of 5-FU to dihydro-5-fluorouracil (FUH_2_). Subsequently, dihydropyrimidinase (DHP, EC 3.5.2.2) catalyzes the hydrolytic ring opening of FUH_2_ to form fluoro-β-ureidopropionic acid (FUPA). Even though DPD and DHP catalysis is reversible, the positive reaction is dominant in vivo [10,11,12,13,14,15]. Lastly, β-ureidopropionase (β-UP, EC 3.5.1.6) catalyzes the hydrolysis of FUPA to fluoro-β-alanine. The three enzymes (DPD, DHP, and β-UP) are encoded by the *DPYD*, *DPYS*, and *UPB1* genes, respectively [16,17,18].

Decreased DPD and DHP enzymatic activities have been linked to genetic polymorphisms identified in patients with severe FP-related toxicities; for each causative polymorphism, the reduction in activity is caused mainly by the substitution or deletion of amino acids [19,20,21]. However, the relationship between β-UP activity and the development of FP-related toxicity is still unknown. To date, the specific effects of previously identified polymorphisms on enzymatic function are largely unknown. Only four *DPYD* variants (c.1905 + 1G > A (IVS14 + 1G > A, *DPYD*2A*); c.1679T > G (*DPYD*13*, p.I560S); c.1129 − 5923C > G /hapB3; and c.2846A > T (p.D949V)) have been characterized as predictive markers for FP-related toxicity in Caucasians [22]. However, significant racial and individual differences in polymorphism location and frequency make it challenging to safely extrapolate the clinical data and institute regional guidelines from one population to another. Thus, it is necessary to further clarify the effects of genetic polymorphisms in an attempt to establish their effect on in vivo enzymatic function. For example, before FP administration, PCR-Restriction Fragment Length Polymorphism (RFLP) analysis, Sanger sequencing, and next-generation sequencing analysis are often used for detecting genetic polymorphisms and establishing patient risk. Moreover, hepatic DPD activity, and thus DPD deficiency incidence, can be predicted by assessing peripheral blood mononuclear cell (PBMC) DPD activity. However, to date, there are no established methods to quantify DHP and β-UP activity clinically.

The most direct method to understand the effect of the genetic polymorphisms of these enzymes on FP pharmacokinetics is to measure metabolite concentrations in blood and urine from subjects with the respective genotypes after FP administration. However, in vivo testing is highly invasive due to continuous blood sampling and poses a considerable risk of FP-related toxicity. Additionally, as the variants of interest are mainly low-frequency polymorphisms, the recruitment of an adequate subject pool to obtain statistically significant data is considerably difficult. While pyrimidine metabolites in blood and urine have been previously quantified to assess enzymatic activity in vivo, these have yielded contradictory results [23,24].

In contrast, in vitro testing using heterologous expression systems has yielded reproducible results using non-invasive methods to facilitate enzymatic activity assessment [25]. Amongst these, several in vitro FP analyses using *Escherichia coli* or mammalian cells have been reported. While other in vitro techniques have been used to evaluate genetic polymorphisms including gene expression profiling, in this review, we focus on the in vitro analysis of the FP-metabolizing enzymes: DPD, DHP, and β-UP, thus providing further information to aid in the application of genetic testing in a clinical setting in light of recent novel insights.

## 2. Dihydropyrimidine Dehydrogenase (DPD)

DPD, the rate-limiting enzyme of the pyrimidine degradation pathway, catalyzes the reduction of 5-FU and uracil to FUH_2_ and dihydrouracil (UH_2_). The DPD gene (*DPYD*) is expressed in most human tissues, but the expression level is highest in the liver and PBMCs [26]. Located on chromosome 1p21, human *DPYD* is comprised of 23 exons and features a 3078 bp open reading frame, encoding a polypeptide containing 1025 amino acid residues [27].

DPD deficiency is an autosomal recessive disorder first reported in a child with neurological symptoms by Bakkeren et al., which was characterized by the accumulation of uracil and thymine in urine, blood, and cerebrospinal fluid [28]. The clinical symptoms include convulsions, autism, microcephaly, growth impairment, and intellectual disability, although asymptomatic cases have also been reported [29,30,31]. The frequency of DPD-deficient patients varies greatly across world populations. While Caucasian frequencies range from 3–5% for partial deficiency and 0.2% for complete deficiency, it is estimated to be extremely rare in Asians [32,33]. In the case of asymptomatic DPD deficiency, there is a considerable risk of FP accumulation during treatment, including 5-FU, which could lead to severe toxicity in patients [34,35,36]. Therefore, it is imperative to diagnose DPD deficiency before chemotherapy administration, even in cases with no prior clinical evidence of this condition.

Of the three metabolic enzymes, *DPYD* is the most studied gene. More than 500 *DPYD* polymorphisms to date have been identified and have been linked to FP-related toxicity in cancer patients [22,37,38,39,40,41,42,43,44,45]. Several of these variants are known to alter amino acid sequence or mRNA splicing, resulting in decreased enzymatic activity. Within the Clinical Pharmacogenetics Implementation Consortium (CPIC) guidelines, three variants (c.85T > C (*DPYD*9A*, p.C29R), c.1627A > G (*DPYD*5*, p.I543V), and c.2194G > A (*DPYD*6*, p.V732I)) are reported to have no effect on enzyme activity [46]. Four variants that cause exon 14 skipping or amino acid substitution (c.1905 + 1G > A (IVS14 + 1G > A, *DPYD*2A*), c.1679T > G (*DPYD*13*, p.I560S), c.1129 − 5923C > G/hapB3, and c.2846A > T (p.D949V)) are designated as having reduced enzymatic function and thus increase the risk of developing toxicity. Similarly, the Dutch Pharmacogenetics Working Group (DPWG) guidelines define these same four variants as risk factors for FP-related toxicity and recommend reducing treatment dosage when a patient possesses one of them [47]. Although *DPYD*9A*, **5*, and **6* are common variants in many ethnic groups, these four risk variants have not yet been identified in Asians [48,49,50].

For most identified *DPYD* variants, except those mentioned above, the effect on DPD activity is unknown, and it is important to clarify the DPD phenotype [51]. The current standard to predict DPD activity measures its enzymatic activity in PBMCs, which correlates with hepatic DPD activity [52,53]. However, this method is not easily implemented in its current form in routine medical care, as it lacks solid evidence of clinical utility. Due to insufficient sensitivity, methods for quantifying pyrimidine metabolites in blood or urine might not identify patients with partial DPD deficiencies [23]. Moreover, additional studies on the clinical validity and utility of these tests are required before implementation can be justified.

In vitro testing is one of the methods used for estimating DPD phenotypes and for the functional analysis of identified non-synonymous variants [54,55,56,57,58,59,60]. Several studies of such tests using *E. coli* or mammalian cell expression systems have been reported (Table 1). Ogura et al. functionally analyzed two variants (G366A and T768K) identified from 150 healthy Japanese volunteers using an *E. coli* expression system [57]. Interestingly, while the G366A mutation produced a decreased intrinsic clearance (*CL_int_*) for 5-FU, reducing DPD activity by 50%, the T768K mutation did not. However, T768K-related activity decreased at a faster rate than that of wild-type DPD, suggesting protein instability. In a subsequent study, Offer et al. expressed 80 non-synonymous variants in HEK293T/c17 cells and measured their enzymatic activities using 5-FU as a substrate [58]. M166V, E828K, K861R, and P1023T exhibited significantly higher activity than wild-type DPD. In contrast, 31 variants, including D949V, exhibited significantly lower activity than wild-type DPD. Elraiyah et al. also analyzed 10 non-synonymous variants identified from 588 Somali and Kenyan individuals using HEK293T/c17 cells [59], in which P86L, P237L, A513V, T793I, V941A, and P1023S exhibited significantly reduced DPD activities. We have characterized 21 DPD allelic variants identified from 1070 Japanese individuals by transient expression in 293FT cells [60]. Among these, 10 (T298M, V313L, V335M, A380V, V434L, V515I, R592W, T768K, H807R, and V826M) showed significantly reduced *CL_int_* values relative to wild-type DPD, and the 5-FU metabolic activity of G926V was practically zero. These reports have yielded consistent results for *DPYD*2A*, which exhibited decreased activity, and for *DPYD*5* (I543V) and **6* (V732I), which exhibited activities that were not considerably different from that of wild-type DPD. In contrast, there are variants such as *DPYD*9A* (C29R) and M166V, whose reported activities differ significantly among previous reports. Ogura et al. and our group found that M166V had a lower activity compared with that of wild-type DPD, while Offer et al. reported a reduction in activity for M166V. The differences in these activities are believed to be due to the differences in assay conditions and cell lines used. Notably, we and Ogura et al. reported DPD variants that were identified almost exclusively in Japanese individuals. Therefore, this raises awareness of the possibility of unidentified rare and relevant ethno-specific variants, which could lead to severe FP-related toxicity.

From a biochemical perspective, human DPD is a flavoprotein containing a single flavin mononucleotide (FMN), a single flavin adenine dinucleotide (FAD), and four iron-sulfur (FeS) clusters. Human DPD consists of five major domains [61,62,63,64]. Domain I (residues 27–172) and domain V (residues 1–26, 848–1025) each contain two FeS clusters. FAD- and nicotinamide adenine dinucleotide phosphate (NADPH)-binding sites are located in domain II (residues 173–286, 442–524) and domain III (residues 287–441), respectively. FMN and the substrate both bind to domain IV (residues 525–847). Human DPD form a dimer, in which electrons from NADPH are transferred to the FeS clusters to catalyze the reduction of bound substrates [65]. Domains II and IV are essential for DPD activity in the structural analysis of variants. Amino acid substitutions that have been observed to affect protein conformation adjacent to the FeS clusters have also caused a significant decrease in enzyme activity.

Henricks et al. described a prediction method using an activity score system and divided *DPYD* alleles into three categories, consisting of fully functional alleles (wild-type; value of 1), reduced activity alleles (c.2846A > T and HapB3; value of 0.5), and nonfunctional alleles (*DPYD*2A* and **13*; value of 0) [66]. Allele values are totaled for a given patient, leading to an individual gene activity score that represents the DPD phenotype of the patient. Moreover, Shrestha et al. developed a *DPYD*-specific variant classifier (*DPYD*-Varifier) using machine learning of in vitro functional data from 156 variants [67]. This model exhibited an accuracy of 85% and outperformed other in silico prediction tools, including PROVEAN, SIFT, and Polyphen-2. In the future, it may be possible to easily predict in vivo DPD activity using machine learning by creating compound databases by gathering detailed information from in vitro analyses. Recently, a list of *DPYD* variants has been added to the Pharmacogene Variation Consortium website (https://www.pharmvar.org/gene/DPYD). It is expected that evidence-based decisions on FP therapeutic regimens and patient-specific dose guidelines could be applied on the basis of an activity score formula, as has been recommended and implemented with other clinically relevant metabolic enzymes.

## 3. Dihydropyrimidinase (DHP)

DHP, as previously mentioned, catalyzes the hydrolytic ring opening of FUH_2_ and UH_2_ and is expressed mainly in the liver and kidneys [15,68]. The human DHP gene (*DPYS)* consists of 10 exons mapped to chromosome 8q22, and features a 1560 bp open reading frame, corresponding to a 519 amino acid protein [17].

DHP deficiency is an autosomal recessive disease characterized by the accumulation of UH_2_ and dihydrothymine (TH_2_) in blood, urine, and cerebrospinal fluid [69]. The clinical phenotype of DHP-deficient patients is highly variable, ranging from asymptomatic to exhibiting symptomatology similar to that of DPD deficiency, including seizures, intellectual disability, growth impairment, and dysmorphic facial features [70,71,72]. To date, 35 genetically confirmed patients with DHP deficiency have been reported [33,73,74,75,76,77]. However, potential asymptomatic deficiencies might be present in a population with a low frequency of DPD deficiencies. In screening 21,200 healthy Japanese infants, Sumi et al. estimated the deficiency frequency to be approximately 1/10,000 [73]. Akai et al. analyzed the *DPYS* coding regions from 183 Japanese individuals, in which the c.349T > C (p.W117R) and c.1001A > G (p.Q334R) variants were identified with an allelic frequency of 0.27% and 1.09%, respectively [78].

To date, multiple studies have reported on the relationship between DPD deficiency and the risk of developing FP-related toxicity. However, there is an increasing awareness that patients with DHP deficiencies are also prone to the development of severe FP-associated toxicity. One such study identified severe FP-related toxicity in a female breast cancer patient with the *DPYS* heterozygous mutation c.833G > A (p.G278D) [21]. We previously reported about a patient with severe capecitabine-associated toxicity and DHP deficiency caused by a compound *DPYS* heterozygous mutation, c.1001A > G (p.Q334R) and c.1393C > T (p.R465X), including a genetic analysis of the patient’s family [79]. Urinary pyrimidine analysis of the patient’s family revealed that the UH_2_/uracil ratio of heterozygous individuals was similar to that of wild-type individuals. Although heterozygous patients are predominantly asymptomatic, severe toxicity might occur during chemotherapy containing FPs, rendering the need for genetic testing before FP administration [80].

It is noteworthy that a sizable number of DHP-deficient patients have been identified in East Asian populations. Hamajima et al. identified a single frameshift mutation and five *DPYS* missense variants in six Japanese patients with dihydropyrimidinuria [17]. Nakajima et al. reported two Chinese pediatric patients with DHP deficiency caused by the compound *DPYS* heterozygous mutation c.1001A > G and c.1443 + 5G > A (exon 8 skipping) [81]. Moreover, Nakajima et al. identified eight variants, including four novel missense mutations and one novel deletion in four DHP-deficient patients [77]. Thus, *DPYS* polymorphisms could emerge as novel pharmacogenomic markers associated with severe FP-related toxicity in diverse global populations.

Recently, in vitro functional characterization of DHP variants using heterologous expression systems, including *E. coli* and mammalian cells, has been reported (Table 2). Van Kuilenburg et al. reported that in the case of 14 variants (L7V, M70T, D81G, G278D, R302Q, L337P, T343A, W360R, V364M, S379R, R412M, R465X, R475X, and R490C) expressed in *E. coli*, the hydrolytic ring opening of radiolabeled UH_2_ was markedly altered [71,76]. Hamajima et al. and Thomas et al. reported that six variants (L7V, T68R, Q334R, W360R, G435R, and R490C) showed lower activities than wild-type DHP in COS-7 and RKO cells expression systems [17,82]. We have characterized 21 DHP variants and wild-type DHP expressed in 293FT cells using UH_2_ and FUH_2_ as substrates [83]. Among these, 13 variants (N16K, T68R, M70T, D81G, G278D, R302Q, L337P, W360R, S379R, G435R, R465X, R475X, and R490C) demonstrated no enzymatic activity, and five variants (W117R, Q334R, T343A, V364M, and R412M) showed significantly lower *CL_int_* values than wild-type DHP. Except for L7V, the results of this study corroborated those of other in vitro studies, suggesting that the specific experimental conditions reflected the in vivo activities of the assayed variants. The divergence observed for L7V might be due to differences in assay conditions, substrate concentrations, or expression systems used.

Hsieh et al. reported that dimer formation is essential for DHP activity [84]. Within the cell, DHP is known to form a tetramer composed of subunits containing two zinc ions each [85,86,87]. Each DHP subunit consists of two domains, a large (β/α)_8_-barrel domain that binds the catalytic dimetal center and a small β-sandwich domain [88]. Each subunit also has two dynamic loops, which act as a lid for the substrate-binding pocket. DHP activity is exerted by the interaction of the C-terminus with the dynamic loop of the neighboring subunit [89,90,91]. We have performed immunoblotting assays of native proteins following blue native polyacrylamide gel electrophoresis and showed that oligomer formation is very important for DHP activity [83]. In the reduced or null-activity variants, the ability of DHP to form oligomers was reduced. The five variants G435R, R465X, R475X, R490C, and R490H introduce mutations in the C-terminus or lead to truncation of the C-terminus, thus affecting oligomer formation and resulting in loss of enzymatic activity. In contrast, the substitutions T68R, M70T, D81G, W117R, M250I, G278D, R302Q, Q334R, L337P, T343A, and R412M exist near the active site of the two dynamic loops, which result in conformational changes in the active site that reduce or eliminate activity. Thus, it has been clarified that changes in DHP activity are associated with amino acid substitutions, as well as changes in oligomer formation and the resulting three-dimensional structure. DHP deficiencies are rarely reported in Caucasians but are highly prevalent in Asians. Thus, we consider that these variants could serve as novel pharmacogenomic markers for the prevention of FP-related toxicity, especially in populations that have a low frequency of symptomatic DPD-deficiency cases.

## 4. β-Ureidopropionase (β-UP)

β-UP catalyzes the irreversible last step, converting FUPA and β-ureidopropionic acid (bUPA) to fluoro-β-alanine and β-alanine, respectively. The human β-UP gene, *UPB1*, is located on chromosome 22q11, contains 10 exons, and features an 1155 bp open reading frame; the gene encodes a polypeptide containing 384 amino acids [18]. Human β-UP activity has been detected predominantly in the liver and kidney [26,92].

β-UP deficiency is an autosomal recessive disease characterized by the accumulation of bUPA and *N*-carbamoyl-β-aminoisobutyric acid (NCBA) in urine, blood, and cerebrospinal fluid [93,94]. To date, 33 genetically confirmed patients with β-UP deficiency have been reported [94,95,96,97,98,99,100]. The clinical phenotype of these patients is highly variable but tends to center around neurological problems. Similar to DHP deficiency, β-UP deficiency is often reported in East Asian populations, including Japan and China. Although it has been reported that severe FP-related toxicity is caused by DPD and DHP deficiencies, little is known about the relationship between β-UP deficiency and FP-related toxicity.

There have been several reports of the in vitro analysis of 13 *UPB1* variants with amino acid substitutions identified in β-UP-deficient patients (Table 3). Van Kuilenburg et al. and Thomas et al. reported that variant A85E expressed in *E. coli* and RKO cells was inactive [93,101]. In a separate study, van Kuilenburg et al., using an *E. coli* expression system, analyzed six β-UP variants (L13S, G235R, R236W, S264R, R326Q, and T359M) that had been previously identified in 16 β-UP-deficient patients, showing a significant reduction or loss of activity in all of them [95]. Nakajima et al. reported that the G31S, E271K, and R326Q variants expressed in HEK293 cells showed profound reductions in activity [97]. Moreover, Nakajima et al. performed native polyacrylamide gel electrophoresis of β-UP expressed in HEK293 cells and showed that octamer formation is necessary for β-UP activity as well as DHP activity. The majority of variants showed a significant reduction in enzymatic activity. However, whether these variants contribute to the development of FP-related toxicity remains unclear.

Fidlerova et al. performed an analysis of the entire *UPB1* coding sequence from 113 Czech cancer patients treated using FP-based chemotherapy [102]. Nine *UPB1* variants were detected in a subpopulation of patients exhibiting severe toxicity, including a novel mutation affecting the coding sequence. An analysis of the effect of *UPB1* variants on FP-related toxicity in the population of all analyzed patients revealed an association between the c.−80C > G (rs2070474) variant and gastrointestinal toxicity. In addition, a strong positive correlation was found between carriers of the homozygous c.−80G variant and the development of severe mucositis. Thomas et al. deduced that the c.−80G variants might alter the potential binding sites of transcription factors, resulting in a statistically non-significant decrease in *UPB1* gene expression in patients who are homozygous for the c.−80G allele. This indicates the possibility that *UPB1* variants have an additive and relatively minor effect on the development of FP-related toxicity compared with that of the *DPYD* and *DPYS* variants.

## 5. Other Considerations

Genetic variations in TYMS, MTHFR, and miR-27a have also been associated with FP-related toxicity. Clinical and preclinical studies have shown the importance of intracellular levels of TYMS, a target for 5-FU involved in DNA repair and synthesis [103], as a determinant of sensitivity to 5-FU treatment. Its overexpression stemming from polymorphic TYMS variations lead to differing response rates to 5-FU therapy [104]. The three most studied TYMS genetic polymorphisms are the variable numbers of tandem repeat (VNTR) polymorphisms comprising 28 bp sequence repeats (rs34743033), rs2853542C > G, and the 3’-untranslated region polymorphism 1494delTTAAAG (rs34489327). These polymorphisms alter gene expression, mRNA stability, or TYMS expression levels, resulting in the development of treatment resistance and toxicity [105,106,107]. MTHFR plays a role in the metabolism of folate and forms the reduced folate cofactor essential for TYMS inhibition by 5-FU. Two non-synonymous variants, c.677C > T (p.A222V, rs1801133) and c.1298A > C (p.E429A, rs1801131), alter intracellular folate distribution and decrease enzymatic activity [105,107]. The micro RNA miR-27a polymorphism (rs895819A > G) has been associated with FP-related toxicity, more so in DPD-deficient patients, as increased miR-27a expression leads to decreased DPD mRNA expression [108,109,110]. To date, however, studies involving these genetic polymorphisms have yielded inconsistent results, and further assessment is needed to assess their clinical utility and potential use as biomarkers.

## 6. Conclusions

FPs are degraded by three metabolic enzymes (DPD, DHP, and β-UP), and a reduction or elimination of their activities leads to severe FP-related toxicity. Therefore, predicting enzymatic activity is critical before the administration of FPs, in which in vitro testing has proven to be a useful complementary method to in vivo testing. This review summarized the findings on the functional characterization of DPD, DHP, and β-UP using in vitro analysis. To date, a large number of DPD variants have been analyzed, giving rise to a significant body of evidence regarding the four most commonly identified risk variants in Caucasians (*DPYD*2A*, *DPYD*13*, c.1129 − 5923C > G/hapB3, and c.2846A > T) that are associated with an increased risk of 5-FU-related toxicity. Additionally, a system for predicting in vivo DPD activity has been developed on the basis of in vitro analysis results. This has provided further evidence that rare DHP variants might be useful predictive biomarkers of FP-related toxicity in populations with low frequencies of DPD deficiency, as is the case for Asians. Notably, β-UP is not known to be associated with FP-related toxicity, although variants with reduced function have been identified. Currently, studies comprising in vivo and in vitro correlation of frequent *DPYD* polymorphisms are advancing applicability as well as underlying the importance of including infrequent *DPYD*, *DPYS*, and *UPB1* variants, as their collective data is insufficient to establish their clinical consequences fully. Additional in vitro and large-scale in vivo studies using standardized methodologies are needed to generate clear evidence for rare variants and verify existing associative studies.

Recently, the underlying mechanisms by which amino acid substitutions alter enzymatic activities by influencing three-dimensional structures have been elucidated; these findings have significant implications toward the interpretation of previously acquired data and how they could be further used to aid clinical decision making for optimal treatments and forewarning the need for alternative chemotherapy regimens. We expect that this report and others related to genetic FP-metabolizing enzyme variants will be useful in the development and further validation of pharmacogenetic testing with the future inclusion of additional biomarkers. In this way, these developments could lead to optimal personalized medicine grounded on genetic polymorphisms.

## Figures and Tables

**Figure 1 jcm-09-02342-f001:**
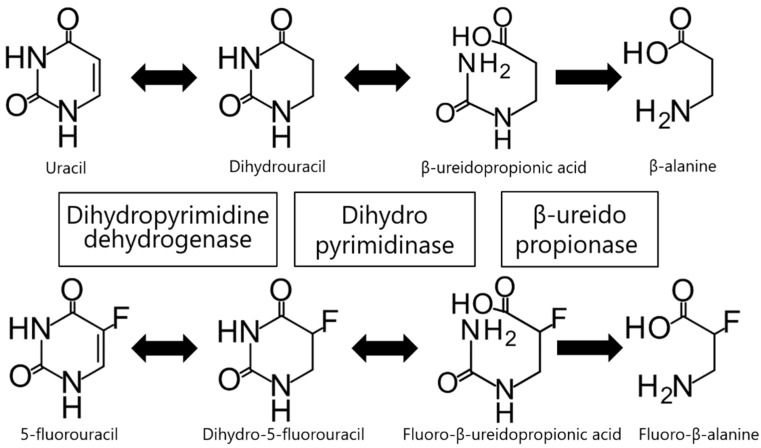
Uracil and 5-fluorouracil degradation pathway. Uracil and 5-fluorouracil are catabolized successively by dihydropyrimidine dehydrogenase, dihydropyrimidinase, and β-ureidopropionase. β-Alanine and fluoro-β-alanine are the final metabolites in this pathway.

**Table 1 jcm-09-02342-t001:** *DPYD* variants reported in in vitro analysis.

dbSNP rsID	PharmVar ID	Location	Nucleotide Change	Amino Acid Substitution	Domain	Expression System	Substrates	Effect	References
rs150036960	PV00901	Exon 2	46C > G	L16V	V	HEK293T/c17	5-FU	Normal function	[58]
rs72549310	PV01042	Exon 2	61C > T	R21X	I	HEK293T/c17	5-FU	No function	[58]
rs80081766	PV01307	Exon 2	62G > A	R21Q	I	HEK293T/c17	5-FU	Normal function	[58]
–	–	Exon 2	74A > G	H25R	I	293FT	5-FU	156% of *CL_int_* ratio	[60]
rs1801265	PV00910	Exon 2	85T > C (*DPYD*9A*)	C29R	I	HEK293T/c17 HEK293 Flp-In	5-FU Thymine	Increased function Decreased function	[54] [55]
rs371587702	PV00962	Exon 3	194C > T	T65M	I	HEK293T/c17	5-FU	Normal function	[58]
–	–	Exon 4	257C > T	P86L	I	HEK293T/c17	5-FU	No function	[59]
rs143986398	PV00887	Exon 4	274C > G	P92A	I	HEK293T/c17	5-FU	Decreased function	[58]
rs72549309	PV01041	Exon 4	295delTCAT (*DPYD*7*)	F100fs	I	HEK293T/c17	5-FU	No function	[58]
rs150385342	PV00902	Exon 4	313G > A	A105T	I	HEK293T/c17	5-FU	Normal function	[58]
–	–	Exon 5	325T > A	Y109N	I	293FT	5-FU	79% of *CL_int_* ratio	[60]
rs141462178	PV00878	Exon 5	343A > G	M115V	I	HEK293T/c17	5-FU	Normal function	[58]
rs200562975	PV00927	Exon 5	451A > G	N151D	I	293FT HEK293T/c17	5-FU 5-FU	107% of *CL_int_* ratio Normal function	[60] [58]
rs2297595	PV0943	Exon 6	496A > G	M166V	I	293FT HEK293T/c17 HEK293 Flp-In	5-FU 5-FU Thymine	77% of *CL_int_* ratio Increased function Decreased function	[60] [58] [55]
rs139834141	PV00871	Exon 6	498G > A	M166I	I	HEK293T/c17	5-FU	Normal function	[58]
rs371792178	–	Exon 6	524C > T	S175L	II	293FT	5-FU	131% of *CL_int_* ratio	[60]
rs115232898	PV00862	Exon 6	557A > G	Y186C	II	HEK293T/c17	5-FU	Decreased function	[58]
rs72549308	PV01040	Exon 6	601A > C	S201R	II	HEK293T/c17	5-FU	No function	[58]
rs72549307	PV01039	Exon 6	632A > G	Y211C	II	HEK293T/c17	5-FU	Decreased function	[58]
rs1801266	PV00911	Exon 7	703C > T (*DPYD*8*)	R235W	II	HEK293T/c17	5-FU	Decreased function	[58]
rs780025995	PV01299	Exon 7	710C > T	P237L	II	HEK293T/c17	5-FU	Decreased function	[59]
rs45589337	PV00984	Exon 8	775A > G	K259E	II	HEK293T/c17	5-FU	Normal function	[58]
rs777220476	PV01275	Exon 9	851G > T	G284V	II	HEK293 Flp-In	Thymine	No function	[56]
rs146356975	PV00895	Exon 9	868A > G	K290E	III	HEK293T/c17	5-FU	Decreased function	[58]
rs143878757	PV00886	Exon 9	893C > T	T298M	III	293FT	5-FU	50% of *CL_int_* ratio	[60]
rs183105782	PV00914	Exon 9	910T > C	Y304H	III	HEK293T/c17	5-FU	Decreased function	[58]
rs150437414	PV00904	Exon 9	929T > C	L310S	III	HEK293T/c17	5-FU	Normal function	[58]
rs145112791	PV00891	Exon 9	934C > T	L312F	III	HEK293T/c17	5-FU	Normal function	[58]
–	–	Exon 9	937G > T	V313L	III	293FT	5-FU	30% of *CL_int_* ratio	[60]
rs201018345	PV00933	Exon 10	967G > A	A323T	III	HEK293T/c17	5-FU	Normal function	[58]
rs72549306	PV01038	Exon 10	1003G > A	V335M	III	293FT	5-FU	47% of *CL_int_* ratio	[60]
rs72549306	PV01037	Exon 10	1003G > T (*DPYD*11*)	V335L	III	HEK293T/c17	5-FU	Normal function	[58]
rs183385770	PV00915	Exon 10	1024G > A	D342N	III	HEK293T/c17	5-FU	Decreased function	[58]
rs190577302	PV00919	Exon 10	1054C > G	L352V	III	HEK293T/c17	5-FU	Decreased function	[58]
rs143154602	PV00882	Exon 10	1057C > T	R353C	III	HEK293T/c17	5-FU	No function	[58]
–	–	Exon 10	1097G > C	G366A	III	293FT *Escherichia coli*	5-FU 5-FU	71% of *CL_int_* ratio 47% of *CL_int_* ratio	[60] [57]
rs72549305	PV01036	Exon 10	1108A > G	I370V	III	HEK293T/c17	5-FU	Normal function	[58]
–	–	Exon 11	1139C > T	A380V	III	293FT	5-FU	33% of *CL_int_* ratio	[60]
–	–	Exon 11	1150A > G	K384E	III	293FT	5-FU	68% of *CL_int_* ratio	[60]
rs78060119	PV01302	Exon 11	1156G > T(*DPYD*12*)	E386X	III	HEK293T/c17	5-FU	No function	[58]
rs140602333	PV00874	Exon 11	1180C > T	R394W	III	HEK293T/c17	5-FU	Normal function	[58]
rs143815742	PV00883	Exon 11	1181G > T	R394L	III	HEK293T/c17	5-FU	Normal function	[58]
rs143815742	PV00884	Exon 11	1181G > A	R394Q	III	HEK293T/c17	5-FU	Normal function	[59]
–	–	Exon 11	1201G > A	G401R	III	HEK293 Flp-In	Thymine	Decreased function	[55]
rs61622928	PV01018	Exon 11	1218G > A	M406I	III	HEK293T/c17 HEK293 Flp-In	5-FU Thymine	Normal function Normal function	[58] [55]
rs200064537	PV00925	Exon 11	1260T > A	N420K	III	HEK293T/c17	5-FU	Normal function	[58]
rs764666241	PV0183	Exon 11	1278G > T	M426I	III	HEK293T/c17	5-FU	Normal function	[58]
rs200693895	PV00931	Exon 11	1280T > C	V427A	III	HEK293 Flp-In	Thymine	Normal function	[56]
rs142512579	PV00880	Exon 11	1294G > A	D432N	III	HEK293T/c17	5-FU	Normal function	[58]
–	–	Exon 11	1300G > C	V434L	III	293FT	5-FU	44% of *CL_int_* ratio	[60]
rs186169810	PV00916	Exon 11	1314T > G	F438L	III	HEK293T/c17	5-FU	Decreased function	[58]
rs72975710	PV01043	Exon 12	1349C > T	A450V	II	HEK293T/c17	5-FU	Normal function	[58]
rs144395748	PV00888	Exon 12	1358C > G	P453R	II	HEK293T/c17	5-FU	Normal function	[58]
rs199549923	PV00921	Exon 12	1403C > A	T468N	II	HEK293T/c17	5-FU	Normal function	[58]
rs72549304	PV01035	Exon 12	1475C > T	S492L	II	HEK293T/c17	5-FU	Decreased function	[58]
rs111858276	PV00857	Exon 12	1484A > G	D495G	II	HEK293T/c17	5-FU	Decreased function	[58]
rs138391898	PV00867	Exon 12	1519G > A	V507I	II	HEK293T/c17	5-FU	Normal function	[58]
rs760663364	PV01150	Exon13	1538C > T	A513V	II	HEK293T/c17	5-FU	Decreased function	[59]
rs148994843	PV00900	Exon 13	1543G > A	V515I	II	293FT HEK293T/c17	5-FU 5-FU	36% of *CL_int_* ratio Normal function	[60] [58]
–	–	Exon 13	1567C > T	L523F	II	HEK293T/c17	5-FU	Normal function	[59]
rs190951787	PV00920	Exon 13	1577C > G	T526S	IV	HEK293T/c17	5-FU	Normal function	[58]
rs1180771326	PV00864	Exon 13	1582A > G	I528V	IV	HEK293T/c17	5-FU	Normal function	[59]
rs1801158	PV00907	Exon 13	1601G > A (*DPYD*4*)	S534N	IV	HEK293T/c17 HEK293 Flp-In	5-FU Thymine	Increased function Decreased function	[54] [55]
rs142619737	–	Exon 13	1615G > C	G539R	IV	HEK293T/c17	5-FU	Normal function	[58]
rs1801159	PV00908	Exon 13	1627A > G (*DPYD*5*)	I543V	IV	293FT HEK293T/c17 HEK293 Flp-In	5-FU 5-FU Thymine	102% of *CL_int_* ratio Normal function Normal function	[60] [54] [55]
rs55886062	PV01000	Exon 13	1679T > G (*DPYD*13*)	I560S	IV	HEK293T/c17	5-FU	Decreased function	[54]
rs201615754	PV00937	Exon 13	1682G > T	R561L	IV	HEK293T/c17	5-FU	Normal function	[58]
rs59086055	PV01015	Exon 14	1774C > T	R592W	IV	293FT HEK293T/c17	5-FU 5-FU	2% of *CL_int_* ratio No function	[60] [58]
rs138616379	PV00869	Exon 14	1775G > A	R592Q	IV	HEK293T/c17	5-FU	Decreased function	[58]
rs145773863	PV00894	Exon 14	1777G > A	G593R	IV	HEK293T/c17	5-FU	No function	[58]
rs147601618	PV00898	Exon 14	1796T > C	M599T	IV	HEK293T/c17	5-FU	Normal function	[58]
Rs72549304	PV01034	Exon 4	1898delC (*DPYD*3*)	P633fs	IV	HEK293T/c17	5-FU	No function	[58]
rs3918289	PV00982	Exon 14	1905C > T/G	N635K	IV	HEK293T/c17	5-FU	Normal function	[58]
rs3918290	PV00983	Intron 14	1905 + 1G > A (*DPYD*2A*)	Exon 14 skipping	IV	HEK293T/c17	5-FU	No function	[54]
rs55971861	PV01003	Exon 15	1906A > C	I636L	IV	HEK293T/c17	5-FU	Normal function	[58]
rs138545885	PV00868	Exon 16	1990G > T	A664S	IV	HEK293T/c17	5-FU	Normal function	[58]
rs137999090	PV00866	Exon 16	2021G > A	G674D	IV	HEK293T/c17	5-FU	No function	[58]
–	–	Exon 17	2096G > C	R699T	IV	HEK293T/c17	5-FU	Normal function	[59]
rs145548112	PV00893	Exon 17	2161G > A	A721T	IV	HEK293T/c17	5-FU	Normal function	[58]
rs146529561	PV00896	Exon 18	2186C > T	A729V	IV	HEK293T/c17	5-FU	Normal function	[58]
rs1801160	PV00909	Exon 18	2194G > A (*DPYD*6*)	V732I	IV	293FT HEK293T/c17 HEK293 Flp-In	5-FU 5-FU Thymine	114% of *CL_int_* ratio Normal function Decreased function	[60] [54] [55]
rs60511679	PV01017	Exon 18	2195T > G	V732G	IV	HEK293T/c17	5-FU	Normal function	[58]
rs112766203	PV00858	Exon 18	2279C > T	T760I	IV	HEK293T/c17	5-FU	Decreased function	[58]
rs56005131	PV01004	Exon 19	2303C > A	T768K	IV	293FT HEK293T/c17 *E. coli*	5-FU 5-FU 5-FU	48% of *CL_int_* ratio Normal function 83% of CLint ratio	[60] [58] [57]
rs199634007	PV00922	Exon 19	2336C > A	T779N	IV	HEK293T/c17	5-FU	Normal function	[58]
rs547099198	PV00994	Exon 19	2378C > T	T793I	IV	HEK293T/c17	5-FU	Decreased function	[59]
–	–	Exon 19	2420A > G	H807R	IV	293FT	5-FU	50% of *CL_int_* ratio	[60]
–	–	Exon 20	2476G > A	V826M	IV	293FT	5-FU	35% of *CL_int_* ratio	[60]
rs200687447	PV00930	Exon 20	2482G > A	E828K	IV	HEK293T/c17	5-FU	Increased function	[58]
rs60139309	PV01016	Exon 20	2582A > G	K861R	V	HEK293T/c17	5-FU	Increased function	[58]
rs201035051	PV00934	Exon 21	2623A > C	K875Q	V	HEK293T/c17	5-FU	Normal function	[58]
rs55674432	PV00996	Exon 21	2639G > T	G880V	V	HEK293T/c17	5-FU	No function	[58]
rs147545709	PV00897	Exon 21	2656C > T	R886C	V	HEK293T/c17	5-FU	Normal function	[58]
rs1801267	PV00912	Exon 21	2657G > A (*DPYD*9B*)	R886H	V	HEK293T/c17	5-FU	Normal function	[58]
rs188052243	PV00918	Exon 21	2678A > G	N893S	V	293FT HEK293T/c17	5-FU 5-FU	61% of *CL_int_* ratio Decreased function	[60] [58]
–	–	Exon 22	2777G > T	G926V	V	293FT	5-FU	No function	[58]
–	–	Exon 22	2822T > C	V941A	V	HEK293T/c17	5-FU	Decreased function	[59]
–	–	Exon 22	2843T > C	I948T	V	HEK293 Flp-In	Thymine	Decreased function	[56]
rs67376798	PV01031	Exon 22	2846A > T	D949V	V	HEK293T/c17 HEK293 Flp-In	5-FU Thymine	Decreased function Decreased function	[58] [55]
rs141044036	PV00876	Exon 22	2872A > G	K958E	V	HEK293T/c17	5-FU	No function	[58]
rs145529148	PV00892	Exon 23	2915A > G	Q972R	V	HEK293T/c17	5-FU	Normal function	[58]
rs72547602	PV01033	Exon 23	2921A > T	D974V	V	HEK293T/c17	5-FU	Normal function	[58]
rs72547601	PV01032	Exon 23	2933A > G	H978R	V	HEK293T/c17	5-FU	No function	[58]
rs61757362	PV01019	Exon 23	2948C > T	T983I	V	HEK293T/c17	5-FU	Decreased function	[58]
rs202144771	PV00941	Exon 23	2977C > T	L993F	V	HEK293T/c17	5-FU	Normal function	[58]
rs139459586	PV00870	Exon 23	2978T > G	L993R	V	HEK293T/c17	5-FU	Normal function	[58]
rs1801268	PV00913	Exon 23	2983G > T (*DPYD*10*)	V995F	V	HEK293T/c17	5-FU	No function	[58]
rs140114515	PV00873	Exon 23	3049G > A	V1017I	V	HEK293T/c17	5-FU	Normal function	[58]
rs148799944	PV00899	Exon 23	3061G > C	V1021L	V	HEK293T/c17	5-FU	Normal function	[58]
rs114096998	PV00860	Exon 23	3067C > A	P1023T	V	HEK293T/c17	5-FU	Normal function	[58]
rs114096998	PV00861	Exon 23	3067C > T	P1023S	V	HEK293T/c17	5-FU	Decreased function	[59]

**Table 2 jcm-09-02342-t002:** *DPYS* variants reported in in vitro analysis.

dbSNP rsID	Location	Nucleotide Change	Amino Acid Substitution	Expression System	Substrates	Effect	References
rs199618701	Exon 1	17G > A	R6Q	293FT	FUH_2_	120% of *CL_int_* ratio	[83]
rs57732538	Exon 1	19C > G	L7V	293FT RKO *E. coli*	FUH_2_UH_2_UH_2_	116% of *CL_int_* ratio 65% of wild-type DHP No function	[83] [82] [76]
rs572241599	Exon 1	48C > G	N16K	293FT	FUH_2_	No function	[83]
–	Exon 1	203C > G	T68R	293FT COS-7	FUH_2_5-bromo-UH_2_	No function 1.5% of wild-type DHP	[83] [17]
rs370718225	Exon 1	209T > C	M70T	293FT *E. coli*	FUH_2_UH_2_	No function No function	[83] [76]
–	Exon 1	242A > G	D81G	293FT *E. coli*	FUH_2_UH_2_	No function No function	[83] [76]
–	Exon 2	349T > C	W117R	293FT	FUH_2_	44% of *CL_int_* ratio	[83]
rs36027551	Exon 3	541C > T	R181W	293FT RKO	FUH_2_UH_2_	110% of *CL_int_* ratio 99% of wild-type DHP	[83] [82]
rs751371011	Exon 4	750G > A	M250I	HEK293	UH_2_	2% of wild-type DHP	[77]
–	Exon 5	833G > A	G278D	293FT *E. coli*	FUH_2_UH_2_	No function No function	[83] [21]
–	Exon 5	884A > G	H295R	HEK293	UH_2_	9.8% of wild-type DHP	[77]
rs200913682	Exon 5	905G > A	R302Q	293FT *E. coli*	FUH_2_UH_2_	No function 3.9% of wild-type DHP	[83] [76]
rs121964923	Exon 6	1001A > G	Q334R	293FT HEK293 COS-7	FUH_2_UH_2_5-bromo-UH_2_	20% of *CL_int_* ratio 9.7% of wild-type DHP 2.5% of wild-type DHP	[83] [77] [17]
rs530911437	Exon 6	1010T > C	L337P	293FT *E. coli*	FUH_2_UH_2_	No function No function	[83] [76]
rs201457190	Exon 6	1027A > G	T343A	293FT *E. coli*	FUH_2_UH_2_	43% of *CL_int_* ratio 49% of wild-type DHP	[83] [76]
rs121964924	Exon 6	1078T > C	W360R	293FT *E. coli* *E. coli* COS-7	FUH_2_UH_2_UH_2_5-bromo-UH_2_	No function No function No function 1.2% of wild-type DHP	[83] [71] [76] [17]
rs138282507	Exon 6	1090G > A	V364M	293FT *E. coli*	FUH_2_UH_2_	8% of *CL_int_* ratio No function	[83] [76]
rs201258823	Exon 7	1137C > A	S379R	293FT *E. coli*	FUH_2_UH_2_	No function 0.20–.9% of wild-type DHP	[83] [76]
rs267606774	Exon 7	1235G > T	R412M	293FT *E. coli*	FUH_2_UH_2_	36% of *CL_int_* ratio No function	[83] [71]
–	Exon 8	1253C > T	T418I	HEK293	UH_2_	64% of wild-type DHP	[77]
rs267606773	Exon 8	1303G > A	G435R	293FT COS-7	FUH_2_5-bromo-UH_2_	No function 5.1% of wild-type DHP	[83] [17]
rs201280871	Exon 8	1393C > T	R465X	293FT *E. coli*	FUH_2_UH_2_	No function No function	[83] [76]
rs61758444	Exon 8	1423C > T	R475X	293FT *E. coli*	FUH_2_UH_2_	No function 0.2–0.9% of wild-type DHP	[83] [76]
rs142574766	Exon 9	1468C > T	R490C	293FT *E. coli* COS-7	FUH_2_UH_2_5-bromo-UH_2_	No function 0.2–0.9% of wild-type DHP 1.7% of wild-type DHP	[83] [76] [17]
Rs189448963	Exon 9	1469G > A	R490H	HEK293	UH_2_	0.3% of wild-type DHP	[77]

**Table 3 jcm-09-02342-t003:** *UPB1* variants identified in β-UP deficient patients.

db SNP rsID	Location	Nucleotide Change	Amino Acid Substitution	Expression System	Substrates	Effect	References
–	Exon 1	c.38T > C	p.L13S	*E. coli*	bUPA	6% of wild-type β-UP	[95]
rs200145797	Exon 1	c.91G > A	p.G31S	HEK293	bUPA	52% of wild-type β-UP	[97]
rs121908066	Exon 2	c.209G > C	p.R70P	No reports of in vitro study			[98]
rs34035085	Exon 2	c.254C > A	p.A85E	*E. coli* RKO	bUPA bUPA	No function 2.7% of wild-type β-UP	[93] [101]
–	Exon 6	c.703G > A	p.G235R	*E. coli*	bUPA	No function	[95]
rs144135211	Exon 6	c.706C > T	p.R236W	*E. coli*	bUPA	No function	[95]
rs145766755	Exon 7	c.792C > A	p.S264R	*E. coli*	bUPA	20% of wild-type β-UP	[95]
–	Exon 7	c.811G > A	p.E271K	HEK293	bUPA	0.7% of wild-type β-UP	[97]
–	Exon 7	c.851G > T	p.C284F	No reports of in vitro study			[99]
rs1375840064	Exon 7	c.853G > A	p.A285T	No reports of in vitro study			[99]
–	Exon 7	c.857T > C	p.I286T	HEK293	bUPA	70% of wild-type β-UP	[97]
rs118163237	Exon 9	c.977G > A	p.R326Q	*E. coli* HEK293	bUPA bUPA	No function 1.3% of wild-type β-UP	[95] [97]
rs369879221	Exon 10	c.1076C > T	p.T359M	*E. coli*	bUPA	No function	[95]

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
