# Peer review of "In Vitro Assessment of Fluoropyrimidine-Metabolizing Enzymes: Dihydropyrimidine Dehydrogenase, Dihydropyrimidinase, and β-Ureidopropionase"

_jcm, 2020, doi:10.3390/jcm9082342_

Round 1

Reviewer 1 Report

This is a well-written review that discuss the pharmacogenomics of FP.

Thi reviewer believe that the authors should also include a section in which they discuss the role of Thymidylate synthase (TYMS) variants in FP clinical outcomes

Author Response

In accordance with the Reviewer's suggestion, we have amended Section 1 and Section 5 as follows:

  1. Introduction

Page 2, Line 43: Genetic polymorphisms of thymidylate synthase (TYMS), methylene tetrahydrofolate reductase (MTHFR), and miR-27a are associated with the development of severe toxicities as well as treatment resistance; however, FP-related toxicity is mainly dependent on FP catabolism.

  1. Other Considerations

Genetic variations in TYMS, MTHFR, and miR-27a have also been associated with FP-related toxicity. Clinical and preclinical studies have shown the importance of intracellular levels of thymidylate synthase, a target for 5-FU involved in DNA repair and synthesis [103], as a determinant of sensitivity to 5-FU treatment. Its overexpression stemming from polymorphic TYMS variations lead to differing response rates to 5-FU therapy [104]. The three most studied TYMS genetic polymorphisms are the variable numbers of tandem repeats (VNTR) polymorphisms comprising of 28-bp sequence repeats (rs34743033), rs2853542C>G, and the 3'-untranslated region polymorphism 1494delTTAAAG (rs34489327). These polymorphisms alter gene expression, mRNA stability, or TYMS expression levels, resulting in the development of treatment resistance and toxicity [105-107]. MTHFR plays a role in the metabolism of folate and forms the reduced folate cofactor essential for TYMS inhibition by 5-FU. Two non-synonymous variants, c.677C>T (p.A222V, rs1801133) and c.1298A>C (p.E429A, rs1801131) alter intracellular folate distribution and decrease enzymatic activity [105,107]. The micro RNA miR-27a polymorphism (rs895819A>G) has been associated with FP-related toxicity, more so in DPD deficient patients, as increased miR-27a expression leads to decreased DPD mRNA expression [108-110]. To date, however, studies involving these genetic polymorphisms have yielded inconsistent results, and further assessment is needed to assess their clinical utility and potential use as biomarkers.

Reviewer 2 Report

I would suggest describing more techniques used for the study of genetic polymorphisms of enzymes in the introduction, both in vivo and in vitro. 

What is the biological meaning of the change in the function of each enzyme as a result of the polymorphism and how can this be related to in vivo studies that are already conducted for some of these variants?

Could the authors give more of a quantitative rather than a qualitative evaluation of the results for each polymorphism? This could mean that if the reduction of the enzymatic function can be somehow quantified then the results could be depicted in the form of a graph in some cases that would be easier for the reader to comprehend and understand the impact of the polymorphisms.

The authors should emphasize more on the correlation of the in vitro with the in vivo results and the importance of in vivo studies in the future at least for the most important polymorphisms.

Author Response

Per the Reviewer's suggestion, we have amended Section 1 as follows:

Page 2, Line 71: For example, before FP administration, PCR-RFLP analysis, Sanger sequencing, and next-generation sequencing analysis are often used for detecting genetic polymorphisms and establishing patient risk. Moreover, hepatic DPD activity, and thus DPD deficiency incidence, can be predicted by assessing peripheral blood mononuclear cells (PBMCs) DPD activity. Although, to date, there are no established methods to quantify DHP and β-UP activity clinically.      

Page 3, Line 86: In contrast, in vitro testing using heterologous expression systems has yielded reproducible results using non-invasive methods to facilitate enzymatic activity assessment [25]. Amongst these, several in vitro FP analyses using Escherichia coli or mammalian cells have been reported. While other in vitro techniques have been used to evaluate genetic polymorphisms including gene expression profiling; in this review, we focus on the in vitro analysis of the FP metabolizing enzymes, DPD, DHP, and β-UP, and thus provide further information to aid in the application of genetic testing in a clinical setting in light of recent novel insights.

Comment 2:

What is the biological meaning of the change in the function of each enzyme as a result of the polymorphism and how can this be related to in vivo studies that are already conducted for some of these variants?

Response: In vivo studies tend only to cover the predominant polymorphisms. In contrast, infrequent and rare variant activity remains unclear both in vivo and in vitro. For the clinically studied variants, functional changes within DPD, DHP, and β-UP can affect in vivo pyrimidine base degradation, depending on the change elicited by the variant, either a decrease or increase in gene expression and resultant enzymatic activity. In the case of enzymatic deficiencies, convulsions, autism, microcephaly, growth impairment, and intellectual disability may occur. Regarding rare and low-frequency variants mentioned within this manuscript, further studies are needed to correlate the in vitro observed effects with possible in vivo consequences, including associative gene studies to elucidate further the effects on cascades and action mechanisms, as well as functional characterization assays using inhibitors and relative combinational substrates.

Comment 3:

Could the authors give more of a quantitative rather than a qualitative evaluation of the results for each polymorphism? This could mean that if the reduction of the enzymatic function can be somehow quantified then the results could be depicted in the form of a graph in some cases that would be easier for the reader to comprehend and understand the impact of the polymorphisms.

Response: We thank the Reviewer for these insightful comments. However, due to discrepancies in the methodologies used across the studies included in this manuscript, it is not possible to qualitatively evaluate the three enzyme polymorphisms. The reports indicate that the substrate concentrations and measurement methods are diverse, and thus, the enzyme activities cannot be compared unequivocally. It is hoped that further in vitro research will unify the results and make it possible to collect consistent data and correlate with in vivo testing and promote clinical applicability.

Comment 4:

The authors should emphasize more on the correlation of the in vitro with the in vivo results and the importance of in vivo studies in the future at least for the most important polymorphisms.

Response: In accordance with the Reviewer's suggestion, we have amended Section 6 as follows:

Page 14, Line 326: Currently, studies comprising in vivo and in vitro correlation of frequent DPYD polymorphisms are advancing applicability as well as underlying the importance of including infrequent DPYD, DPYS, and UPB1 variants, as their collective data is insufficient to establish their clinical consequences fully. Additional in vitro and large-scale in vivo studies using standardized methodologies are needed to generate clear evidence for rare variants and verify existing associative studies.

Reviewer 3 Report

Hishinuma and his colleagues in this review report recapped the results of in vitro analyses of genetic variants of 3 important enzymes (DPD, DHP, and β-UP) in the metabolism of Fluoropyrimidine drugs (FPs) related to clinical usefulness  for pharmacogenomics (PGx) application

Though FPs such as 5-FU and  its oral prodrugs (tegafur, capecitabine, and doxifluridine) are the cornerstone of treatment for various types of cancer including gastrointestinal, head and neck, and breast cancers, they are generally characterized by a large inter-individual pharmacokinetic variability that can cause unexpected toxicity or ineffective treatment. The balance of efficacy and toxicity is critical and the imbalance can have devastating effects on patients. Scientific evidence proves that interindividual variability in drug response is extensive and, in this variability, genetic factors may involve crucial role. To this end, this review manuscript highlights the importance of depth analysis of genetic polymorphism predicting in vitro enzyme activity for potential applications of PGx in clinical practice in the context of tailored and personalized medicine.  Also, in addition to genetic variants it has been proposed that the application of PGx tests could be more appropriate and efficient if integrated with routine therapeutic drug monitoring (TDM) in order to establish well-defined phenotype-genotype correlation, particularly for the drugs with a very narrow therapeutic range such as 5_FU leading to potential serious ADRs/toxicity outside that range

Over all the manuscript is very well written and the introduction provides a good, generalized background of the topic that quickly gives the reader an appreciation of the applications of the technique. Moreover, the topic and the data are very interesting. Certainly this study will contribute much to the literature.

Very Good job

Thank you 

Author Response

We sincerely appreciate the positive feedback and encouraging comments from the Reviewer.